# Local Differential Privacy-Preserving Spectral Clustering for General Graphs

**Sayan Mukherjee**\*  *sayan@phys.s.u-tokyo.ac.jp*
*The University of Tokyo*

**Vorapong Suppakitpaisarn**\*  *vorapong@is.s.u-tokyo.ac.jp*
*The University of Tokyo*

**Reviewed on OpenReview:** *https://openreview.net/forum?id=zo5b60AuAH*

## Abstract

Spectral clustering is a widely used algorithm to find clusters in networks. Several researchers have studied the stability of spectral clustering under local differential privacy with the additional assumption that the underlying networks are generated from the stochastic block model (SBM). However, we argue that this assumption is too restrictive since social networks do not originate from the SBM. Thus, we delve into an analysis for general graphs in this work. Our primary focus is the edge flipping method – a common technique for protecting local differential privacy. We show that, when the edges of an $n$-vertex graph satisfying some reasonable well-clustering assumptions are flipped with a probability of $O(\log n/n)$, the clustering outcomes are largely consistent. Empirical tests further corroborate these theoretical findings. Conversely, although clustering outcomes have been stable for non-sparse and well-clustered graphs produced from the SBM, we show that in general, spectral clustering may yield highly erratic results on certain well-clustered graphs when the flipping probability is $\omega(\log n/n)$. This indicates that the best privacy budget obtainable for general graphs is $\Theta(\log n)$.

## 1 Introduction

As the demand for trustworthy artificial intelligence grows, the need to protect user privacy becomes more crucial. Several methods have been proposed to address this concern. Among these, differential privacy is the most common. Introduced by Dwork (2008), differential privacy measures the amount of privacy a system leaks by using a metric called the privacy budget. This method involves corrupting users' information, then processing the corrupted data to obtain statistical conclusions while still maintaining privacy. Developing algorithms that can accurately provide statistical conclusions from the corrupted information is a topic of interest among many researchers such as Zhu et al. (2017). One of the advantages of the differential privacy notion is that the information revealed from the users' sensitive information is quantified by a term called *privacy budget*. We say that an algorithm protects users' information when its privacy budget is small.

In this work, we are interested in a variant of differential privacy called local differential privacy introduced by Kasiviswanathan et al. (2011). Unlike traditional differential privacy, local differential privacy does not allow the collection of all users' information before it is corrupted. Instead, it requires users to corrupt their data at their local devices before sending them to central servers. This ensures that users' information is not leaked during transmission. As discussed by Erlingsson et al. (2014) and Apple's Differential Privacy Team (2017), local differential privacy is used by companies for their services.

We focus on algorithms for social networks. In a social network, each user is represented by a node, and their relationships with other users are represented by edges. The relationship or edge information is usually

---

\*Equal contribution.

sensitive because one might not want others to know that they have a relationship with some particular individuals. One of the most common techniques for protecting user relationship information under the local differential privacy notion is randomized response or edge flipping, which is a technique considered in Warner (1965), Mangat (1994), and Wang et al. (2016). In this technique, before sending their adjacency vector (which represents their friend list) to the central server, each bit in the adjacency vector is flipped with a specified probability $p$. We obtain local differential privacy with the budget of $\Theta(\log 1/p)$ by the flipping.

Because of the simplicity of randomized response, it has been extensively used as a part of many differentially private graph algorithms Imola et al. (2021); Hillebrand et al. (2025). These include graph clustering algorithms such as Ji et al. (2020); Mohamed et al. (2022); Fu et al. (2023). One of the most widely used and scalable graph clustering algorithms – spectral clustering – has also received a lot of attention in this context. We are particularly interested in the combination of randomized response and spectral clustering, because it has been shown in Peng & Yoshida (2020) that spectral clustering is robust against random edge removal. It could also be robust against edge flipping. Many analyses such as Hehir et al. (2022) have been recently done for this combination. However, all of these analyses assume that the input social networks are generated from the stochastic block model (SBM).

## 1.1 Our Contribution

We argue that assuming that the input graph is generated from the SBM is too restrictive. Thus, in this study, we consider the robustness of spectral clustering for general graphs. In what follows, let $G$ be an $n$-vertex input graph. Our main contribution of Section 3 can be summarized by the following theorem:

**Theorem 1.1** (Informal version of Theorem 3.1)**.** *Let $G'$ be obtained from $G$ via the edge flipping mechanism with probability $p = O(\log n/n)$. Then, under some reasonable assumptions, the number of vertices misclassified by the spectral clustering algorithm by running it on $G'$ instead of $G$ is $O(\eta(G) \cdot n)$ with probability $1 - o(1)$, where $\eta(G)$ is the spectral robustness defined in 2.1.*

We shall see later that $\eta(G)$ is a constant much smaller than 1 for well-clustered graphs. Theorem 1.1 implies that:

$$\text{Spectral clustering is robust against edge flipping or the randomized response} \atop \text{method with probability } p = O(\log n/n), \text{ or privacy budget } \epsilon = \Omega(\log n). \tag{1.1}$$

The privacy budget $\Omega(\log n)$ can be considered too large in many applications, which might limit the contribution of this work. However, this large privacy budget has also been considered in some previous works on differentially private graph clustering such as Mohamed et al. (2022). As the authors prove that a constant privacy budget can be achieved for non-sparse, well-clustered graphs generated by SBM, one might anticipate a similar outcome for general non-sparse, well-clustered graphs. Howeover, in Section 4, we show the following negative result that the edge flipping mechanism cannot achieve better privacy:

$$\text{There is a family of non-sparse and well-clustered graphs for which edge flipping} \atop \text{with probability } p = \omega(\log n/n) \text{ massively changes the sparsest cut.} \tag{1.2}$$

Our findings are depicted in Figure 1.1. This figure focuses on a segment of the Facebook network in the Stanford network analysis project (SNAP) described in Leskovec & Mcauley (2012), as sourced from the "0.edges" file, called as Facebook0 in this paper. We have modified each relationship in this network with a 0.005 probability of flipping, and the figure displays the network both before and after these changes. Observations from the figure reveal significant additions and removals of edges, suggesting that we can protect user information via edge-flipping. For instance, the connectivity (or degree) of node 86 is notably higher after the flipping. However, it is worth noting that the clustering characteristics remain largely unchanged before and after the network modification.

One of the results of Mohamed et al. (2022) proves (1.1), assuming that the input social networks are generated from the SBM. However, we found that no graphs in practice satisfy properties of SBMs. For example, if a graph is generated from the SBM, its degree distribution should follow a mixture of two binomials. However, we found that none of the publicly available datasets at SNAP (Leskovec & Krevl,

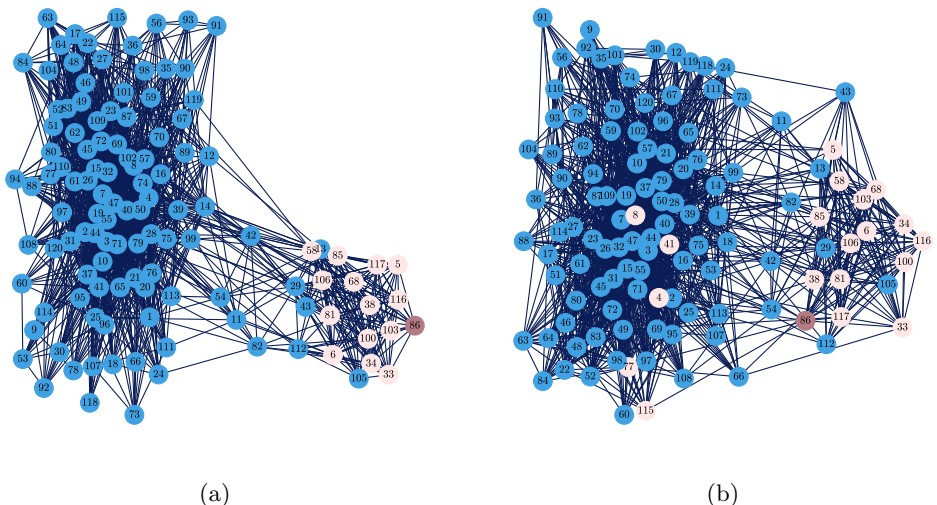

(a) (b)

Figure 1.1: A part of the Facebook network detailed in Leskovec & Mcauley (2012) before and after flipping edges with a probability of 0.005. The neighborhood (colored light pink) of node 86 changes a lot after the flipping.

2014) follow this degree distribution. This motivates us to make much weaker assumptions in this work. The only two assumptions we require are: 1) the social network has a sufficient cluster structure and 2) its maximum degree is sufficiently large. The SBM's considered by Mohamed et al. (2022) also satisfy these assumptions (see Remark 2.13).

We use some ideas from the work of Peng & Yoshida (2020), who have studied the sensitivity of spectral clustering algorithms. However, their work focuses on scenarios where each edge is *removed* with a specific probability. In contrast, local differential privacy not only removes edges but also adds edges to social networks. Furthermore, the number of edges added is often much greater than those removed, especially for sparse networks. Thus, while our proof structure generally follows the outline of their proof, the core components (such as Lemma 3.2, Lemma 3.3 and Theorem 4.1) are original and require careful probabilistic arguments.

**Remark 1.2.** *For many readers, it may seem counter-intuitive that the privacy budget increases with the number of users, given that differential privacy tends to be more effective with larger databases. This can be explained by considering the nature of the data being protected. In relational databases or general graph differential privacy, there are n pieces of information to protect. However, for local edge differential privacy, the protection extends to $O(n^2)$ bits of edge information.*

**Remark 1.3.** *Spectral clustering analysis under local differential privacy is a relatively recent area of exploration. However, there is a substantial body of work on graph clustering with differential privacy, as evidenced by studies like Mohamed et al. (2022) and Wang et al. (2013). Notably, a recent study by Chen et al. (2023) provides both upper and lower limits for privacy budgets pertaining to non-sparse graphs generated from the SBM.*

## 2 Preliminaries

### 2.1 Notation

**Edge-subsets.** We assume that $G = (V, E(G))$ is a graph of $n$ vertices. Also, assume that the set of nodes $V$ is $\{1, \ldots, n\}$. For any subset $F \subseteq \binom{V}{2}$, we denote by $G \triangle F$ the graph $(V, E(G) \triangle F)$. By $F \sim_p \binom{V}{2}$, we mean that $F$ is a random subset of $\binom{V}{2}$ such that each subset is taken with probability $p$.

**Cuts.** For a subset $S \subseteq V$ of vertices, we denote by $\bar{S}$ the complement set $V \setminus S$. Further, given two subsets $A, B \subseteq V$ with $A \cap B = \varnothing$, let $e_G(A, B)$ denote the number of edges of $G$ with one endpoint in $A$ and one in $B$. For any two sets of nodes $S, S' \subseteq V$, $d_{\text{size}}(S, S')$ is given by $d_{\text{size}}(S, S') = \min\left(|S \triangle S'| + |\bar{S} \triangle \bar{S'}|, |S \triangle \bar{S'}| + |\bar{S} \triangle S'|\right)$. As $|S \triangle T| = |\bar{S} \triangle \bar{T}|$, we can equivalently write $d_{\text{size}}(S, S') = 2|S \triangle S'|$. A cut $(S, \bar{S})$ is similar to $(S', \bar{S'})$ if $d_{\text{size}}(S, S')$ is small.

**Spectral Graph Theory.** Any $n \times n$ real symmetric matrix $A$ has $n$ real eigenvalues. We denote the $i$-th smallest eigenvalue of $A$ as $\lambda_i(A)$, i.e. $\lambda_1(A) \leq \lambda_2(A) \leq \cdots \leq \lambda_n(A)$. For any graph $G$, the Laplacian matrix $L_G$ is given by $D_G - A_G$, where $D_G$ is the diagonal degree matrix with $(D_G)_{ii} = \deg_G(i)$ and $A_G$ is the adjacency matrix of $G$.

In this work, we relate the performance of spectral clustering under the edge-flipping mechanism via the following quantity:

**Definition 2.1** (Spectral robustness). *We define the spectral robustness of the graph $G$ as $\eta(G) := \frac{\Delta(G)\lambda_2(L_G)}{\lambda_3(L_G)^2}$, where $\Delta(G)$ denotes the maximum degree of any vertex of $G$.*

## 2.2 Edge Differential Privacy under Randomized Response

The notion of $\varepsilon$-edge differential privacy is defined as follows:

**Definition 2.2** ($\varepsilon$-edge differential privacy; Nissim et al. (2007)). *Let $G$ be a social network and let $Y$ be a randomized mechanism that outputs $Y(G)$ from the social network $G$. For any $\varepsilon > 0$, any possible output of the mechanism $Y$ denoted by $y$, and any two social networks $G^{(1)} = (V, E^{(1)}(G))$ and $G^{(2)} = (V, E^{(2)}(G))$ that differ by one edge, we say that $Y$ is $\varepsilon$-edge differentially private if $e^{-\varepsilon} \leq \frac{\Pr[Y(G^{(1)})=y]}{\Pr[Y(G^{(2)})=y]} \leq e^{\varepsilon}$. We refer to the value of $\varepsilon$ as the privacy budget of $Y$.*

Intuitively, a lower value of $\varepsilon$ results in better privacy protection. In this research, for $0 \leq p \leq 0.5$, we investigate a randomized mechanism $Y_p$ that seeks to generate a result highly similar to spectral clustering outcomes, using randomized response. The mechanism $Y_p$ is defined as $Y_p = \mathcal{SC} \circ \mathcal{F}_p$, where $\mathcal{F}_p$ represents a randomized function that modifies the relationship between each node pair with a probability of $p$, and $\mathcal{SC}$ is a function for computing spectral clustering. In other words, the randomized mechanism performs spectral clustering on $G \triangle F$, in which $(u, v) \in F$ with a probability of $p$ for every $u, v \in V$. The following theorem is shown in Wang et al. (2016).

**Theorem 2.3** (Wang et al. (2016)). *The publication $Y_p$ is $\varepsilon$-edge differentially private whenever $\frac{1-p}{p} \leq e^{\varepsilon}$.*

Theorem 2.3 implies that $Y_p$ is $\varepsilon$-edge differential private for $\varepsilon \geq \ln(1 - p) - \ln p$. When $p$ is small, we have $\ln(1 - p) \approx 0$, and therefore the privacy budget of the publication $Y_p$ is $\Omega(\log 1/p)$.

## 2.3 Spectral Clustering

For a graph $G$, the general goal of clustering techniques is to find a good cut $(S, \bar{S})$ such that $e_G(S, \bar{S})$ is small, and most of the edges of $G$ are either concentrated in $S$ or $\bar{S}$. In order to avoid trivial cuts (for example where $S$ comprises of a single vertex), it is customary to instead define the *cut-ratio* $\alpha_G(S) = \frac{e_G(S,\bar{S})}{|S||\bar{S}|}$ and find cuts that minimize $\alpha_G(S)$ (see Wei & Cheng (1989); Hagen & Kahng (1992)). $\alpha(G) = \min_{\varnothing \subsetneq S \subsetneq V} \alpha_G(S)$ is defined as the cut-ratio of $G$. Unless otherwise specified, we shall denote by $S^*$ the cut that achieves $\alpha_G(S^*) = \alpha(G)$.

Another widely used way of defining the cut-ratio is $\alpha'_G(S) = \frac{e_G(S,\bar{S})}{\min(|S|,|\bar{S}|)}$. This definition is used in Peng & Yoshida (2020), Guattery & Miller (1995), and Kwok et al. (2013). We observe that these two definitions are related:

**Lemma 2.4.** $\frac{1}{2} \cdot n\alpha_G(S) \leq \alpha'_G(S) \leq n\alpha_G(S)$.

*Proof.* For the left side of the inequality, note that $\frac{n}{2} \cdot \alpha_G(S) = \frac{|S|+|\bar{S}|}{2} \cdot \frac{e_G(S,\bar{S})}{|S||\bar{S}|} = \frac{1}{2}\left(\frac{e_G(S,\bar{S})}{|S|} + \frac{e_G(S,\bar{S})}{|\bar{S}|}\right) \leq$
$\max\left\{\frac{e_G(S,\bar{S})}{|S|}, \frac{e_G(S,\bar{S})}{|\bar{S}|}\right\} = \alpha'_G(S)$. On the other hand, $\alpha'_G(S) = \alpha_G(S) \cdot \max(|S|,|\bar{S}|) \leq n \cdot \alpha_G(S)$. □

Lemma 2.4 will be useful in converting results formulated using $\alpha'_G$ to those using our cut-ratio $\alpha_G$.

Now we describe the spectral clustering algorithm. Spectral clustering uses the eigenvalues and eigenvectors of $L_G$ to compute a cut of $S$. Let us denote by $\mathcal{SC}_2$ the following algorithm that clusters a given graph $G$ into two clusters:

- Compute the second smallest eigenvector $\vec{v} = [\mathsf{v}_1, \ldots, \mathsf{v}_n]^\intercal$ of $L_G$ using the Lanczos algorithm or an approximation method, such as the one proposed in Adil & Saranurak (2024). Let $v_1, \ldots, v_n$ be distinct nodes of $V$ such that $\mathsf{v}_{v_1} \leq \cdots \leq \mathsf{v}_{v_n}$.

- Return the cut $(S, \bar{S})$, where $S = \{v_1, \ldots, v_{i_0}\}$ and $i_0 = \arg\min\limits_{1 \leq i \leq n} \alpha_G(v_1, \ldots, v_i)$.

The cut-ratio of $G$ can be quantified very precisely via the famous Cheeger's inequality.

**Lemma 2.5** (Cheeger's Inequality; Cheeger (1971); Alon (1986)). $\lambda_2(L_G) \leq n\alpha(G) \leq \sqrt{8\Delta(G)\lambda_2(L_G)}$.

We shall also use the following improvement of Lemma 2.5:

**Lemma 2.6** (Improved Cheeger Inequality; Kwok et al. (2013)). *Let $\mathcal{SC}_2(G)$ denote the cut given by the spectral clustering algorithm. Then, $\alpha_G(\mathcal{SC}_2(G)) \leq O\left(\frac{\lambda_2(L_G)\Delta(G)^{1/2}}{n\lambda_3(L_G)^{1/2}}\right)$.*

Lemma 2.5 and 2.6 give us a way of quantifying the quality of the cut output by $\mathcal{SC}_2$ in terms of the cut-ratio of $G$. Indeed, as $\frac{\lambda_2(L_G)}{n} \leq \alpha(G)$,

$$\alpha_G(\mathcal{SC}_2(G)) \leq O\left(\frac{\Delta(G)^{1/2}}{\lambda_3(L_G)^{1/2}}\right) \cdot \alpha(G). \tag{2.1}$$

Let $S^*$ be the cut of $G$ with the smallest cut-ratio. While equation (2.1) can be interpreted as a measure of how close $\mathcal{SC}_2(G)$ is with $S^*$, we shall need stability results from Peng & Yoshida (2020) to precisely bound $d_{\text{size}}(\mathcal{SC}_2(G), S^*)$.

**Lemma 2.7** (Lemma 3.5 of Peng et al. (2015)). *Let $G = (V, E)$ be any graph with optimal min-cut $S^*$. Then, for any $\rho \geq 1$, if $S \subseteq V$ satisfies $\alpha'_G(S) \leq \rho \cdot \alpha_G(S^*)$, then $d_{size}(S, S^*) = \Theta\left(\frac{\lambda_2(L_G)\Delta(G)^{1/2}}{\lambda_3(L_G)^{3/2}} \cdot \rho\right) \cdot n$.*

The following lemma is directly obtained from Lemma 2.7.

**Lemma 2.8** (Stability of min-cut). *Let $G = (V, E)$ be any graph with optimal min-cut $S^*$. Then, for any $\rho \geq 1$, if $S \subseteq V$ satisfies $\alpha_G(S) \leq \rho \cdot \alpha_G(S^*)$, then $d_{size}(S, S^*) \leq O\left(\frac{\lambda_2(L_G)\Delta(G)^{1/2}}{\lambda_3(L_G)^{3/2}} \cdot \rho\right) \cdot n$.*

*Proof.* Observe that by Lemma 2.4, $\alpha_G(S) \leq \rho \cdot \alpha_G(S^*)$ implies $\alpha'_G(S) \leq 2\rho \cdot \alpha'_G(S^*)$. This lemma then follows from a direct application of Lemma 3.5 of Peng & Yoshida (2020). □

### 2.4 Concentration Inequalities

We also require some concentration inequalities for random variables, which we present here.

**Lemma 2.9** (Hoeffding's inequality; Hoeffding (1963)). *Let $X_1, \ldots, X_n$ be independent random variables such that $a_i \leq X_i \leq b_i$ almost surely. If $S = X_1 + \cdots + X_n$, then we have $\Pr[S \leq \mathbb{E}(S) - t] \leq \exp\left(-2t^2 / \sum_{i=1}^n (b_i - a_i)^2\right)$.*

**Lemma 2.10** (Chernoff bound; Mitzenmacher & Upfal (2017)). *For a binomial random variable $X$ with mean $\mu$ and $t > 0$, we have $\Pr[X \geq \mu + t] \leq \exp\left(-\frac{t^2}{2\mu+t}\right)$.*

**Lemma 2.11** (Weyl's Inequality; Weyl (1912)). *For any real symmetric matrices $M$ and $H$ $|\lambda_i(M + H) - \lambda_i(M)| \leq \|H\|_2$, where $\|H\|_2$ denotes the spectral norm of $H$.*

### 2.5 Assumptions

In order to demonstrate the robustness of spectral clustering, we require assumptions on the social network $G$ and the probability $p$ of edge flipping. Recall that $F \sim_p \binom{V}{2}$ is the set of vertex pairs to be flipped.

**Assumption 2.12.** *We assume the following:*

1.  $p < \log n / 10n$,
2.  *(a)* $\Delta(G) \geq 10 \log n \lambda_3(L_G)$,    *(b)* $\lambda_2(L_G) \geq 1/10$,    *(c)* $\eta(G) := \frac{\lambda_2(L_G)\Delta(G)}{\lambda_3(L_G)^2}$ *is small*,
    *(d)* $\lambda_3(L_G) \geq 10 \log n$,
3.  *Let the minimum cuts of $G$ and $G \triangle F$ be $(S^*, \overline{S^*})$ and $(S_F^*, \overline{S_F^*})$, respectively. Then each of $|S^*|, |\overline{S^*}|, |S_F^*|, |\overline{S_F^*}|$ have size at least $n/10$.*

*Plausibility of Assumption 2.12*:

The first assumption can be justified by our discussion in Section 2.2, where we note that to achieve a privacy budget of $O(\log n)$, $p$ should be $\Omega(1/n)$. We further note that, if $G$ is a sparse social network with $O(n)$ edges and $p \gg \log n/n$, then as $\mathbb{E}(|F|) = \Omega(n \log n)$, $G \triangle F$ will have too much noise, and would become close to the Erdős-Rényi random graph $F \sim \mathcal{G}(n, p)$. Spectral algorithms cannot perform well for these graphs. For example, it is shown in Chung & Radcliffe (2011) that the eigenvalues of the normalized Laplacian $\mathcal{L}_F$ are close to those of the expected values. A quick calculation shows that the second and third eigenvalues of $\mathbb{E}(\mathcal{L}_F)$ are both equal (and close to 1), implying the inefficiency of spectral clustering algorithms on $\mathcal{G}(n, p)$ for $p$ asymptotically larger than $\log n/n$.

On the other hand, one may think that values of $p$ larger than $\log n/n$, for example $p = \Omega(1)$ is achievable by the edge flipping mechanism if the input graph $G$ is not sparse. However, there are two issues with this: firstly, social networks are not dense in practice. We found that all publicly available social networks in SNAP (Leskovec & Krevl, 2014) are sparse. Secondly, we demonstrate in Section 4, a well-clustered non-sparse graph, whose sparsest cut changes drastically when introducing noise $p = \omega(\log n/n)$.

The second assumption derives from usual properties of social networks. Recall that we have $0 = \lambda_1(L_G) \leq \lambda_2(L_G) \leq \cdots \leq \lambda_n(L_G) < 2\Delta(G)$. This assumption asserts that there are big gaps between $\lambda_2(L_G)$, $\lambda_3(L_G)$ and $\Delta(G)$.

First, we note that most social networks that we encounter in practice, have super-nodes (nodes of degree $\Omega(n)$) (Barabási & Albert, 1999), justifying our assumption (a).

Assumption (b) ensures that $G$ is well-connected: note that disconnected graphs have $\lambda_2(L_G) = 0$ and graphs that have small edge-separators have a small $\lambda_2(L_G)$. We also make a note here that we can relax this assumption to any constant threshold $\lambda_2(L_G) \geq \Theta(1)$, by changing the value of $\gamma_0$ used in Section 3.3. Conversely, the counterexample provided in Section 4 has connectivity of $O(1/n)$, implying that for the stability of $\mathcal{SC}_2$, the algebraic connectivity of $G$ cannot be too small.

Finally, (c) ensures that there is a gap between $\lambda_3(L_G)$ and $\lambda_2(L_G)$, which ensures that the graph has a good bi-cluster structure, which lets $\mathcal{SC}_2$ find good clusters in $G$. The large gap between $\lambda_2(L_G)$ and $\lambda_3(L_G)$ is usually assumed in several works on spectral clustering such as Peng et al. (2015); Peng & Yoshida (2020) since the large gap implies that the graph is well-clustered.

Observe that using inequalities (a), (b) and (c), we can deduce that $\lambda_3(L_G) = \frac{\lambda_2(L_G)\Delta(G)}{\lambda_3(L_G)\eta(G)} \geq \frac{\log n}{\eta(G)}$, which implies our assumption of (d).

Our final assumption stems from the fact that usually social networks admit linearly sized clusters, and also we are usually interested in detecting clusters of larger size via the definition of the cut ratio $\alpha(G)$, for example. Moreover, our analysis in Equation 3.6 suggests that $|S_F^*|$ and $|\overline{S_F^*}|$ are close to the output $|S_F|$ and $|\overline{S_F}|$ of $\mathcal{SC}_2$, which is designed to output balanced cuts. This justifies our assumption of $|S_F^*|, |\overline{S_F^*}| \geq 0.1n$.

**Remark 2.13.** *When dealing with SBM's with probabilities $p = a \log n/n$ inside each cluster and $q = b \log n/n$ between, the work of Deng et al. (2021) proves that $\lambda_2(L_G) \leq O(b \log n)$ and $\lambda_3(L_G) \geq b \log n$. Therefore, these SBM's satisfy Assumption 2.12, and the result for Randomized Response in Mohamed et al. (2022) follows from our work.*

## 3 Main Theorem

We restate and prove a formal version of Theorem 1.1 in this section.

**Theorem 3.1.** *Let $G = (V, E)$ be a graph and $p$ satisfy Assumption 2.12. Let $F \sim_p \binom{V}{2}$. Then, with probability at least $1 - 5n^{-8/5}$,*

$$d_{size}\left(\mathcal{SC}_2(G), \mathcal{SC}_2(G \triangle F)\right) = O(\eta(G) \cdot n).$$

**Proof Structure.** Suppose $S^*$ and $S_F^*$ are the optimum min-cuts of $G$ and $G \triangle F$. Denote by $S$ and $S_F$ the outputs of $\mathcal{SC}_2$ on $G$ and $G \triangle F$, respectively.

The key idea is to bound $d_{\text{size}}(S, S_F)$ using triangle inequality:

$$d_{\text{size}}(S, S_F) \le d_{\text{size}}(S, S^*) + d_{\text{size}}(S^*, S_F^*) + d_{\text{size}}(S_F^*, S_F). \tag{3.1}$$

We bound each of the terms in their own subsection below. Observe that by Equations (3.2), (3.6) and (3.8), we obtain

$$d_{\text{size}}(S, S_F) \le O\left(\frac{\lambda_2(L_G)\Delta(G)}{\lambda_3(L_G)^2}\right) \cdot n = O(\eta(G) \cdot n)$$

with probability at least $1 - 4n^{-21/11} - n^{-8/5} \ge 1 - 5n^{-8/5}$, completing the proof. In the remainder of this section, we bound each term appearing in the right side of Equation (3.1).

### 3.1 The term $d_{\text{size}}(S, S^*)$.

An upper bound on this term is a direct corollary of Cheeger's inequality and stability: observe that Lemma 2.8 and (2.1) give us

$$d_{\text{size}}(S, S^*) \le O\left(\frac{\lambda_2(L_G)\Delta(G)^{1/2}}{\lambda_3(L_G)^{3/2}} \cdot \frac{\Delta(G)^{1/2}}{\lambda_3(L_G)^{1/2}}\right) \cdot n = O\left(\frac{\lambda_2(L_G)\Delta(G)}{\lambda_3(L_G)^2}\right) \cdot n \tag{3.2}$$

### 3.2 The term $d_{\text{size}}(S_F^*, S_F)$.

First, we describe a lemma to compare the eigenvalues and maximum degrees of $G \triangle F$ and $G$.

**Lemma 3.2.** *Let $G$ have $n$ vertices, and $F \sim \mathcal{G}(n, p)$. Under Assumption 2.12, with probability at least $1 - 3n^{-21/11}$, all of the following hold:*

*(a) $\lambda_2(L_{G \triangle F}) \le \lambda_2(L_G)$,    (b) $\lambda_3(L_{G \triangle F}) \ge \lambda_3(L_G)/10$,    (c) $\Delta(G \triangle F) \le 2\Delta(G)$.*

*Proof. Part (a).* By monotonicity of $\lambda_2$, $\lambda_2(L_{G \triangle F}) \le \lambda_2(L_{G \cup F})$. As $\lambda_2(L_{G \cup F}) \le \lambda_2(L_G) + \lambda_2(L_{F \setminus G})$, and $p < \log n / 10n$, $F$ (and hence $F \setminus G$) is almost surely disconnected Erdős et al. (1960), implying $\lambda_2(L_{F \setminus G}) = 0$. Hence, we have $\lambda_2(L_{G \triangle F}) \le \lambda_2(L_{G \cup F}) \le \lambda_2(L_G)$. ∎

*Part (b).* For this part, we shall use Weyl's Inequality as follows: suppose $F_1 = F \setminus G$ and $F_2 = G \cap F$ be subgraphs of $F$ on the vertex set $V(G)$. By additivity of the Laplacian, $L_{G \triangle F} - L_G = L_{F_1} - L_{F_2}$. Now as $\|A\|_2 = \max_{x \in \mathbb{R}^n} x^\mathsf{T} A x$ for any symmetric $n \times n$ matrix $A$, which implies

$$\|L_{G \triangle F} - L_G\|_2 = \max_{x \in \mathbb{R}^n} |x^\mathsf{T} L_{F_1} x - x^\mathsf{T} L_{F_2} x| \le \max_{x \in \mathbb{R}^n} x^\mathsf{T} L_F x = \lambda_n(L_F) \le 2\Delta(F).$$

By the union bound, note that for any $v \in V(G)$,

$$\Pr[\Delta(F) > \frac{9}{2} \log n] \le n \cdot \Pr[\deg_F(v) > \frac{9}{2} \log n] \le n \cdot \Pr\left[\deg_F(v) - p(n-1) > 4 \log n\right]. \tag{3.3}$$

Using the Chernoff bound, the probability in (3.3) is at most

$$n \cdot \exp\left(-\frac{16(\log n)^2}{2(n-1)p + 4 \log n}\right) < n \cdot \exp\left(-\frac{16(\log n)^2}{\frac{11}{2} \log n}\right) = n \cdot \exp\left(-\frac{32}{11} \log n\right) = n^{-21/11}, \tag{3.4}$$

Thus $\|L_{G\triangle F} - L_G\|_2 \leq 9\log n$ holds with probability at least $1 - n^{-21/11}$. By Weyl's inequality and Assumption 2.12(2),

$$\lambda_3(L_G) - \lambda_3(L_{G\triangle F}) \leq 9\log n \leq \frac{9}{10}\lambda_3(L_G),$$

finishing the proof of (b). ∎

*Part (c).* Observe that for every vertex $v \in V(G)$, we have

$$\deg_{G\triangle F}(v) - \deg_G(v) \leq \deg_F(v) \leq \Delta(F).$$

Hence,

$$\Pr\left[\deg_{G\triangle F}(v) > \deg_G(v) + \Delta(G)\right] \leq \Pr\left[\Delta(F) > \Delta(G)\right] \leq \Pr\left[\Delta(F) > 10\log n\right].$$

By a similar calculation to (3.3) and (3.4), we conclude that $\deg_{G\triangle F}(v) > \deg_G(v) + \Delta(G)$ holds with probability at most $n^{-4}$. Again, by the union bound, with probability at least $1 - n^{-3}$, we have

$$\deg_{G\triangle F}(v) \leq \deg_G(v) + \Delta(G) \text{ for all } v \in V(G). \tag{3.5}$$

Taking the maximum of (3.5) over all $v$, we see that (c) holds with probability at least $1 - n^{-3}$, which is greater than $1 - n^{-21/11}$. ∎

As the assertions of (a), (b), (c) each hold with probability at least $1 - n^{-21/11}$, all of them simultaneously hold with probability at least $1 - 3n^{-21/11}$, completing our proof of Lemma 3.2. □

Now, observe that by the same argument as (3.2) in addition with Lemma 3.2, we get that with probability at least $1 - 3n^{-21/11}$,

$$d_{\text{size}}(S_F^*, S_F) \leq O\left(\frac{\lambda_2(L_{G\triangle F})\Delta(G\triangle F)}{\lambda_3(L_{G\triangle F})^2}\right) \cdot n = O\left(\frac{\lambda_2(L_G)\Delta(G)}{\lambda_3(L_G)^2}\right) \cdot n \tag{3.6}$$

## 3.3 The term $d_{\text{size}}(S^*, S_F^*)$.

For the remainder of this section, let $\gamma_0$ be given by

$$\gamma_0 := 200\sqrt{\Delta(G)/\lambda_3(L_G)} > 200\sqrt{10\log n}.$$

In order to bound $d_{\text{size}}(S^*, S_F^*)$, we require the following rather technical lemma.

**Lemma 3.3.** *Let $S^*$ denote the minimum cut of $G$ and $S_F^*$ denote the minimum cut of $G\triangle F$. Suppose $n/2 \geq |S^*|, |S_F^*| \geq \epsilon n$ for some $1/2 > \epsilon > 0$. Further, suppose $\alpha_G(S_F^*) \geq \gamma_0\alpha_G(S^*)$. Then,*

$$\Pr\left(\gamma_0\alpha_{G\triangle F}(S_F^*) - \alpha_{G\triangle F}(S^*) < 0\right) < \exp\left(-\frac{4\left(\gamma_0^2 - 1\right)^2}{25\gamma_0^2} \cdot \alpha_G(S^*)^2\epsilon^2 n^2\right). \tag{3.7}$$

As the proof is involved, we defer it to the end of this section.

First, we demonstrate the bound on $d_{\text{size}}(S^*, S_F^*)$ using Lemma 3.3. We consider two cases:

- **Case 1.** $\alpha_G(S_F^*) \leq \gamma_0\alpha_G(S^*)$: In this case, Lemma 2.8 directly gives us $d_{\text{size}}(S^*, S_F^*) \leq O\left(\frac{\gamma_0\lambda_2(L_G)\Delta(G)^{1/2}}{\lambda_3(L_G)^{3/2}}\right) \cdot n = O\left(\frac{\lambda_2(L_G)\Delta(G)}{\lambda_3(L_G)^2}\right) \cdot n.$

- **Case 2.** $\alpha_G(S_F^*) > \gamma_0\alpha_G(S^*)$: In this case, setting $\epsilon = 1/10$ in Lemma 3.3, we note that the probability that $\alpha_{G\triangle F}(S^*) > \gamma_0\alpha_{G\triangle F}(S_F^*)$ is at most:

$$\exp\left(-\frac{(2\gamma_0^2 - 2)^2\alpha_G(S^*)^2 n^2}{2500\gamma_0^2}\right) < \exp\left(-\frac{\gamma_0^2}{2500} \cdot (\alpha_G(S^*) \cdot n)^2\right) < \exp\left(-160\log n \cdot \lambda_2(L_G)^2\right)$$

$$< \exp\left(-160\log n \cdot \frac{1}{100}\right) = n^{-8/5}$$

The last line follows from Assumption 2.12(2). Hence, with probability at least $1 - n^{-8/5}$, $\alpha_{G\triangle F}(S^*) \leq \gamma_0 \alpha_{G\triangle F}(S_F^*)$ holds. By Lemma 2.8, this implies $d_{\text{size}}(S^*, S_F^*) \leq O\left(\frac{\gamma_0 \lambda_2(L_{G\triangle F})\Delta(G\triangle F)^{1/2}}{\lambda_3(L_{G\triangle F})^{3/2}}\right) \cdot n$. Together with Lemma 3.2, we obtain that with probability at least $1 - n^{-8/5} - 3n^{-21/11}$,

$$d_{\text{size}}(S^*, S_F^*) \leq O\left(\frac{\Delta(G)^{1/2}}{\lambda_3(L_G)^{1/2}} \cdot \frac{\lambda_2(L_{G\triangle F})\Delta(G\triangle F)^{1/2}}{\lambda_3(L_{G\triangle F})^{3/2}}\right) \cdot n = O\left(\frac{\lambda_2(L_G)\Delta(G)}{\lambda_3(L_G)^2}\right) \cdot n, \qquad (3.8)$$

finishing our upper bound on $d_{\text{size}}(S^*, S_F^*)$. ∎

We now present our proof of Lemma 3.3.

*Proof.* (Lemma 3.3). The main idea behind the proof is as follows: first, we show that Lemma 3.3 holds with $S_F^*$ replaced with any fixed subset $A$. Then, we use the fact that

$$\Pr\left(\gamma_0 \alpha_{G\triangle F}(S_F^*) < \alpha_{G\triangle F}(S^*)\right) = \Pr\left(\gamma_0 \alpha_{G\triangle F}(S_F^*) < \alpha_{G\triangle F}(S^*) \mid \alpha_G(S_F^*) > \gamma_0 \alpha_G(S^*)\right)$$

$$= \sum_{A:\alpha_G(A)>\gamma_0\alpha_G(S^*)} \Pr(S_F^* = A) \cdot \Pr\left(\gamma_0 \alpha_{G\triangle F}(S_F^*) < \alpha_{G\triangle F}(S^*) \mid S_F^* = A\right)$$

$$\leq \max_{A:\alpha_G(A)>\gamma_0\alpha_G(S^*)} \Pr\left(\gamma_0 \alpha_{G\triangle F}(A) < \alpha_{G\triangle F}(S^*)\right), \tag{3.9}$$

as $\sum_A \Pr(S_F^* = A) = 1$.

Now, we bound $\Pr\left(\gamma_0 \alpha_{G\triangle F}(A) < \alpha_{G\triangle F}(S^*)\right)$ for any fixed $A$.

**Claim 3.4.** *Let $S^*$ denote the minimum cut of $G$. Suppose $\frac{n}{2} \geq |S^*| \geq \epsilon n$ for some $\frac{1}{2} > \epsilon > 0$. Then, for any $\gamma > 1$ and $\frac{n}{2} \geq |A| \geq \epsilon n$,*

$$\Pr\left(\gamma \alpha_{G\triangle F}(A) - \alpha_{G\triangle F}(S^*) < 0\right) < \exp\left(-\frac{4\left(\gamma \alpha_G(A) - \alpha_G(S^*)\right)^2}{25\gamma^2} \cdot \epsilon^2 n^2\right). \tag{3.10}$$

*Proof of Claim 3.4.* Let $Y_A := \gamma \alpha_{G\triangle F}(A) - \alpha_{G\triangle F}(S^*)$. We wish to show that $Y_A \geq 0$ with high probability.

For any tuple $(x, y) \in V \times V$, define $X_{(x,y)}$ as the boolean random variable such that $X_{(x,y)} = 1$ if $xy \in E(G\triangle F)$ and $X_{(x,y)} = 0$ otherwise. As $X_{(x,y)} = X_{(y,x)}$, we abuse notation and write $X_{xy}$ as a shorthand for both these variables. Note that $X_{xy}$ are all mutually independent, and

$$\mathbb{E}(X_{(x,y)}) = \Pr(xy \in E(G\triangle F)) = \begin{cases} p, & \text{if } e \notin E(G), \\ 1-p, & \text{if } e \in E(G). \end{cases} \tag{3.11}$$

Further, for any subset $A \subseteq V$, by definition

$$\alpha_{G\triangle F}(A) = \frac{e_{G\triangle F}(A, \overline{A})}{|A||\overline{A}|} = \frac{1}{|A||\overline{A}|} \cdot \sum_{(x,y)\in A\times\overline{A}} X_{(x,y)},$$

Which, by (3.11), implies

$$\mathbb{E}(\alpha_{G\triangle F}(A)) = \frac{1}{|A||\overline{A}|} \cdot \left(\sum_{e\in E_G(A,\overline{A})} \mathbb{E}(X_e) + \sum_{e\in A\times\overline{A}\setminus E_G(A,\overline{A})} \mathbb{E}(X_e)\right)$$

$$= \frac{1}{|A||\overline{A}|} \cdot \left(e_G(A, \overline{A}) \cdot (1-p) + |A||\overline{A}| \cdot p - e_G(A, \overline{A}) \cdot p\right)$$

$$= (1 - 2p) \cdot \alpha_G(A) + p. \tag{3.12}$$

Let $\mu$ denote the expectation of $Y_A$. By linearity and (3.12),

$$\mu = \mathbb{E}(Y_A) = (1 - 2p) \cdot (\gamma \alpha_G(A) - \alpha_G(S^*)) + p \cdot (\gamma - 1) > \frac{4}{5} \cdot (\gamma \alpha_G(A) - \alpha_G(S^*)), \qquad (3.13)$$

As $\gamma > 1$ and $p < 1/10$. We also have $\mu > 0$, and $\Pr(Y_A < 0) = \Pr(Y_A - \mu < -\mu)$. Now we shall use Hoeffding's inequality to provide an upper bound on $\Pr(Y_A < 0)$. To that end, $Y_A$ has to be rewritten as a sum of independent random variables. However,

$$Y_A = \frac{\gamma}{|A||\overline{A}|} \cdot \sum_{e \in A \times \overline{A}} X_e - \frac{1}{|S^*||\overline{S^*}|} \cdot \sum_{e \in S^* \times \overline{S^*}} X_e. \qquad (3.14)$$

As the two summations in $Y_A$ have overlapping terms, we separate them as follows. Let $Z_1 = S^* \setminus A$, $Z_2 = S^* \cap A$, $Z_3 = A \setminus S^*$, $Z_4 = \overline{S^* \cup A}$. Observe then,

$$\begin{aligned} S^* \times \overline{S^*} &= (Z_1 \times Z_3) \sqcup (Z_1 \times Z_4) \sqcup (Z_2 \times Z_3) \sqcup (Z_2 \times Z_4) \\ A \times \overline{A} &= (Z_3 \times Z_1) \sqcup (Z_3 \times Z_4) \sqcup (Z_2 \times Z_1) \sqcup (Z_2 \times Z_4) \end{aligned} \qquad (3.15)$$

This lets us break each sum in (3.14) into four parts, and using $X_{(x,y)} = X_{(y,x)}$, we can write $Y$ as

$$Y_A = \sum_{e \in (Z_1 \times Z_3) \sqcup (Z_2 \times Z_4)} \left( \frac{\gamma}{|A||\overline{A}|} - \frac{1}{|S^*||\overline{S^*}|} \right) X_e + \sum_{e \in (Z_1 \times Z_4) \sqcup (Z_2 \times Z_3)} \frac{\gamma X_e}{|A||\overline{A}|} - \sum_{e \in (Z_3 \times Z_4) \sqcup (Z_1 \times Z_2)} \frac{X_e}{|S^*||\overline{S^*}|}. \qquad (3.16)$$

Note that all summands in (3.16) are independent of each other. For simplicity, let us denote $z_i := |Z_i|$ for $i = 1, \ldots, 4$.

Since $-|c| \leq cX_e \leq |c|$ for any constant $c \in \mathbb{R}$, we can use Hoeffding's inequality to get $\Pr(Y < 0) = \Pr(Y_A - \mu < -\mu) \leq \exp(-\frac{2\mu^2}{D})$, where $D = 4(z_1 z_3 + z_2 z_4) \left( \frac{\gamma}{(z_2 + z_3)(z_1 + z_4)} - \frac{1}{(z_1 + z_2)(z_3 + z_4)} \right)^2 + \frac{4\gamma^2(z_1 z_4 + z_2 z_3)}{(z_2 + z_3)^2(z_1 + z_4)^2} + \frac{4(z_3 z_4 + z_1 z_2)}{(z_1 + z_2)^2(z_3 + z_4)^2}$. After some calculations, this leads to

$$D = \frac{4\gamma^2(z_1 + z_2)(z_3 + z_4)}{(z_2 + z_3)^2(z_1 + z_4)^2} + \frac{4(z_2 + z_3)(z_1 + z_4)}{(z_1 + z_2)^2(z_3 + z_4)^2} - \frac{8\gamma(z_1 z_3 + z_2 z_4)}{(z_1 + z_2)(z_3 + z_4)(z_2 + z_3)(z_1 + z_4)} \qquad (3.17)$$

$$< 4\gamma^2 \left( \frac{(z_1 + z_2)(z_3 + z_4)}{(z_2 + z_3)^2(z_1 + z_4)^2} + \frac{(z_2 + z_3)(z_1 + z_4)}{(z_1 + z_2)^2(z_3 + z_4)^2} \right) \qquad (3.18)$$

$$= 4\gamma^2 \left( \frac{|S^*||\overline{S^*}|}{|A|^2|\overline{A}|^2} + \frac{|A||\overline{A}|}{|S^*|^2|\overline{S^*}|^2} \right) \leq 4\gamma^2 \cdot 2 \cdot \frac{n^2/4}{\epsilon^2 n^4/4} = \frac{8\gamma^2}{\epsilon^2 n^2}. \qquad (3.19)$$

Here (3.19) follows from the fact that $n^2/4 \geq |S^*||\overline{S^*}|, |A||\overline{A}| \geq \epsilon n \cdot n/2$. Therefore, in conjunction with (3.13), we obtain $\Pr(Y_A < 0) \leq \exp\left( -\frac{2\mu^2}{D} \right) < \exp\left( -\frac{4(\gamma \alpha_G(A) - \alpha_G(S^*))^2}{25\gamma^2} \cdot \epsilon^2 n^2 \right)$, as desired. $\blacksquare$

Now we return to our proof of Lemma 3.3. For any set $A \subseteq V$ with $\frac{n}{2} \geq |A| \geq \epsilon n$ and $\alpha_G(A) > \gamma_0 \alpha_G(S^*)$, we have $\Pr(\gamma_0 \alpha_{G \triangle F}(A) < \alpha_{G \triangle F}(S^*)) < \exp\left( -\frac{4(\gamma_0 \alpha_G(A) - \alpha_G(S^*))^2}{25\gamma_0^2} \cdot \epsilon^2 n^2 \right) \leq \exp\left( -\frac{4(\gamma_0^2 - 1)^2}{25\gamma_0^2} \cdot \alpha_G(S^*)^2 \epsilon^2 n^2 \right)$. When we plug this back into Equation (3.9), it gives our desired bound.

$$\square$$

# 4   Instability of spectral clustering when $p = \omega(\log n / n)$

We now construct a family of random graphs $G$, whose sparsest cut (in expectation) drastically changes under edge flipping with $p = \omega(\log n / n)$. The formal construction is given in Theorem 4.1. In the construction, we denote by $G \cong \mathcal{G}(\mathsf{n}, \mathsf{p})$ a random graph $G$ generated according to the Erdős–Rényi model with $\mathsf{n}$ nodes and connection probability $\mathsf{p}$. Denote $F \cong \mathcal{G}(n, p)$, and recall that the result of the edge flipping is $G \triangle F$. Furthermore, we consider the matrices whose entries are the expected values of the corresponding entries in $L_G$ and $L_{G \triangle F}$. These matrices are denoted by $\mathbb{E}[L_G]$ and $\mathbb{E}[L_{G \triangle F}]$, respectively.

**Theorem 4.1.** *Let $1/2 > \beta > 1/10$ be a constant. Let $G$ be a graph on $n$ vertices with vertex set $A \sqcup B \sqcup C$, where $|A| = \beta n$, $|B| = |C| = (1-\beta)n/2$. Suppose the induced subgraphs of $G$ on $A$, $B$ and $C$ satisfy $G[A] \cong \mathcal{G}(\beta n, 400\log^2 n/n)$, $G[B] \cong \mathcal{G}(n, \log n/n)$ and $G[C] \cong \mathcal{G}((1-\beta)n/2, 400\log^2 n/n)$. Further, suppose that every pair of vertices $(x,y)$ with $x \in B$ and $y \in C$ are adjacent with a probability of $20\log n/n$. Finally, let any pair of vertices $(x,y)$ with $x \in A$ and $y \in B \cup C$ are adjacent with probability $1/10n$. A visual representation of this construction is shown in Figure 4.1 (left). Then, the following holds:*

1. *$G$ satisfies Assumption 2.12 in expectation. In other words, the expected maximum degree of $G$, denoted by $\overline{\Delta}$, satisfies $\overline{\Delta} \geq 10\log n\lambda_3(\mathbb{E}[L_G])$, the value of $\lambda_2(\mathbb{E}[L_G])$ is larger than $1/10$, the value of $\frac{\lambda_2(\mathbb{E}[L_G])\overline{\Delta}(G)}{\lambda_3(\mathbb{E}[L_G])^2}$ is small, and $\lambda_3(\mathbb{E}[L_G]) \geq 10\log n$.*

2. *When $p = \omega(\log n/n)$, $\lim_{n\to\infty} \frac{\lambda_2(\mathbb{E}[L_{G\triangle F}])}{\lambda_3(\mathbb{E}[L_{G\triangle F}])} = 1$.*

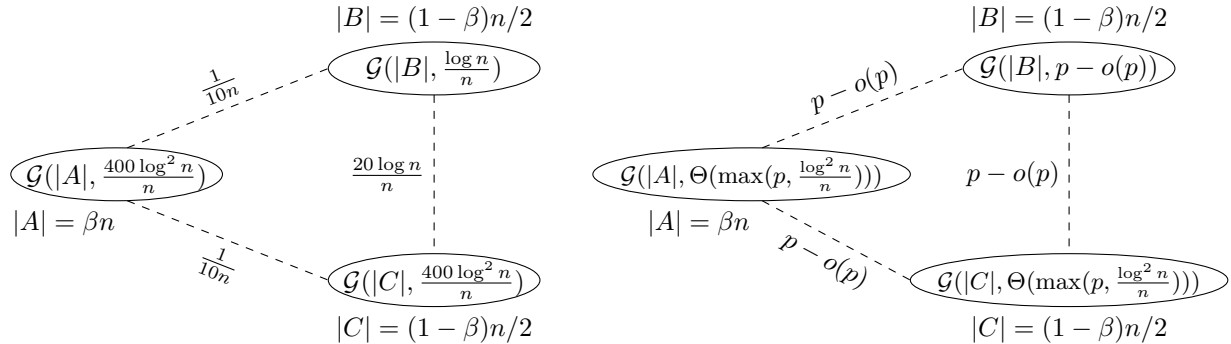

Figure 4.1: The graph $G$ (left) and the graph $G\triangle F$ where $F \cong \mathcal{G}(n,p)$ (right). Dashed lines represent probabilistic edges between the parts $A$, $B$ and $C$.

*Proof of (1).* It is straightforward to see that the expected maximum degree of $G$ satisfies

$$\overline{\Delta} = \left((1-\beta)\frac{n}{2} - 1\right) \cdot \frac{400\log^2 n}{n} + (1-\beta)\frac{n}{2} \cdot \frac{20\log n}{n} + \beta n \cdot \frac{1}{10n} = 200(1-\beta)\log^2 n + O(\log n). \quad (4.1)$$

Next, we observe that $\mathbb{E}[L_G]$ is a block matrix with blocks corresponding to $A$, $B$, $C$. The first three eigenvectors of $\mathbb{E}[L_G]$ would therefore take constant values on the blocks $A$, $B$ and $C$. The normalized eigenvector of $\lambda_1(\mathbb{E}[L_G]) = 0$ is constant on all three blocks, that of $\lambda_2(\mathbb{E}[L_G])$ is constant on $A$ and $B \cup C$ (since the density of edges between $B$ and $C$ is larger than between $A$ and $B \cup C$), and that of $\lambda_3(\mathbb{E}[L_G])$ can take different values on each blocks. Suppose the eigenvector of $\lambda_2(\mathbb{E}[L_G])$ takes the value $x_2$ on $A$ and $y_2$ on $B \cup C$, then, by the fact that the sum of all elements of the eigenvector is 0, we must have $\beta x_2 + (1-\beta)y_2 = 0$. Also, by the fact that the sum square of all elements of the eigenvector is 1, we must have $\beta n x_2^2 + (1-\beta)n y_2^2 = 1$. This implies that $x_2 = \pm\sqrt{\frac{1-\beta}{\beta n}}$ and $y_2 = \mp\sqrt{\frac{\beta}{(1-\beta)n}}$. Let the second eigenvector of $\mathbb{E}[L_G]$ be $(\nu_{2,i})_{i\in V}$. Since $\lambda_2(\mathbb{E}[L_G]) = \sum_{\{i,j\}\in E}(\nu_{2,i} - \nu_{2,j})^2$, we obtain that:

$$\lambda_2(\mathbb{E}[L_G]) = \frac{\beta n \cdot (1-\beta)n}{10n} \cdot (x_2 - y_2)^2 = \frac{\beta(1-\beta)}{10} \cdot \frac{4}{\beta(1-\beta)n} = \frac{4}{10}. \quad (4.2)$$

Now, as the eigenvector of $\lambda_3(\mathbb{E}[L_G])$ will take values of, say, $x_3$ on $A$, $y_3$ on $B$ and $z_3$ on $C$, then, because the sum of all elements of the eigenvector is 0, they satisfy $\beta x_3 + (1-\beta)y_3/2 + (1-\beta)z_3/2 = 0$. Because the sum square of all elements of the eigenvector is 1, they satisfy $\beta n x_3^2 + (1-\beta)n y_3^2/2 + (1-\beta)n z_3^2/2 = 1$. Additionally, since the second and the third eigenvectors are orthogonal, we obtain that $\beta x_2 \cdot x_3 + \frac{1-\beta}{2}y_2 \cdot y_3 + \frac{1-\beta}{2}y_2 \cdot z_3 = 0$. Since $\beta x_2 + (1-\beta)y_2 = 0$ and $\beta x_2 = -(1-\beta)y_2$, we obtain that $x_3 + y_3/2 + z_3/2 = 0$. By solving the three

equations, we obtain that $x_3 = 0$, $y_3 = \pm\frac{1}{\sqrt{(1-\beta)n}}$, $z_3 = \mp\frac{1}{\sqrt{(1-\beta)n}}$. Let the third eigenvector of $\mathbb{E}[L_G]$ be $(\nu_{3,i})_{i \in V}$. Since $\lambda_3(\mathbb{E}[L_G]) = \sum_{\{i,j\} \in E}(\nu_{3,i} - \nu_{3,j})^2$:

$$\lambda_3(\mathbb{E}[L_G]) = \frac{\beta(1-\beta)n^2}{10n} \cdot \frac{1}{(1-\beta)n} + \frac{(1-\beta)^2 n^2 \cdot 20\log n}{4n} \cdot \frac{4}{(1-\beta)n} = 20(1-\beta)\log n + O(1). \quad (4.3)$$

For these values, we notice that $\eta(G) = \frac{200(1-\beta)\log^2 n \cdot 4}{400(1-\beta)^2 \log^2 n \cdot 10} = \frac{1}{5(1-\beta)}$ is small, thus $G$ satisfies Assumption 2.12 in expectation. $\qquad\square$

*Proof of (2).* Since $G \triangle F$ is obtained from randomly flipping adjacencies with probability $p$, observe that $\mathbb{E}(A_{G \triangle F}) = pJ + (1-2p)\mathbb{E}(A_G)$, where $J$ is the all-ones matrix. Note that the average degrees of the nodes of $G \triangle F$ are the entries of $\mathbb{E}(A_{G \triangle F})\mathbf{1}$ where $\mathbf{1}$ is the all-ones vector. Thus, $\mathbb{E}(A_{G \triangle F})\mathbf{1} = pn\mathbf{1} + (1-2p)\mathbb{E}(A_G)\mathbf{1}$. For the diagonal degree matrices, this implies that $\mathbb{E}(D_{G \triangle F}) = \mathrm{diag}(\mathbb{E}(A_{G \triangle F})\mathbf{1}) = pnI + (1-2p)\mathbb{E}(D_G)$. In terms of the average Laplacian matrices, this gives us the relation

$$\mathbb{E}[L_{G \triangle F}] = pnI + (1-2p)\mathbb{E}(D_G) - pJ - (1-2p)\mathbb{E}(A_G) = pn\left(I - \frac{J}{n}\right) + (1-2p)\mathbb{E}[L_G]. \quad (4.4)$$

Let $\{\nu_1 = \mathbf{1}, \nu_2, \ldots, \nu_n\}$ be the eigenvectors of $\mathbb{E}[L_G]$. Observe that $\mathbb{E}[L_{G \triangle F}]\mathbf{1} = 0$, thus $\nu_1 = \mathbf{1}$ is clearly an eigenvector of $\mathbb{E}[L_{G \triangle F}]$. On the other hand, when $i \neq 1$, by orthogonality we have that $J\nu_i = 0$, and thus,

$$\mathbb{E}[L_{G \triangle F}]\nu_i = pn\nu_i + (1-2p)\mathbb{E}[L_G]\nu_i = (pn + (1-2p)\lambda_i(\mathbb{E}[L_G]))\nu_i. \quad (4.5)$$

Equation (4.5) implies that $\mathbb{E}[L_{G \triangle F}]$ and $\mathbb{E}[L_G]$ share the same eigenvectors, and moreover, since we assume that $p \leq 0.5$ in this work, we obtain that for $i \neq 1$,

$$\lambda_i(\mathbb{E}[L_{G \triangle F}]) = pn + (1-2p)\lambda_i(\mathbb{E}[L_G]). \quad (4.6)$$

Now, since $np = \omega(\log n) = \omega(\lambda_3(\mathbb{E}[L_G]))$, we obtain

$$\lim_{n \to \infty} \frac{\lambda_2(\mathbb{E}[L_{G \triangle F}])}{\lambda_3(\mathbb{E}[L_{G \triangle F}])} = \lim_{n \to \infty} \frac{pn + (1-2p)\lambda_2(\mathbb{E}[L_G])}{pn + (1-2p)\lambda_3(\mathbb{E}[L_G])} = 1. \quad (4.7)$$

$$\square$$

Since the Laplacian matrices $L_G$ and $L_{G \triangle F}$ closely resemble their expectations $\mathbb{E}[L_G]$ and $\mathbb{E}[L_{G \triangle F}]$ as discussed in Chung & Radcliffe (2011); Lu & Peng (2013), Theorem 4.1 implies that the graph $G$ satisfies Assumption 2.12. Moreover, the ratio $\frac{\lambda_2(L_{G \triangle F})}{\lambda_3(L_{G \triangle F})}$ is close to one, indicating that spectral clustering is unlikely to yield accurate clustering results with high probability.

## 5 Experiments

We conduct experiments on real social networks to verify our theoretical results. In this work, we mainly use the network called "Social circles: Facebook" obtained from the Stanford network analysis project (SNAP), detailed in Leskovec & Mcauley (2012). We found that spectral clustering cannot find plausible results for some of those networks, due to the fact that there are many connected components, and also small sets of nodes with only one to two edges to the rest of the graph. Those small sets usually form a cluster in the outcomes of spectral clustering, which makes the outcomes undesirable. We therefore decided to eliminate all small node sets that have at most 10 outgoing edges.

We examine the graphs defined in the files "0.edges" and "1684.edges." We call the graphs as Facebook0 and Facebook1684. After removing nodes of small degree, there are $n = 120$ left in the first graph and $n = 574$ left in the second. As illustrated in Figure 1.1, the social network Facebook0 is composed of two clusters, both of which are quite sizable. Alternatively, the social network Facebook1684 is divided into three clusters. The first is a small yellow cluster including nodes 0–15, followed by a bigger cluster of yellow nodes, and

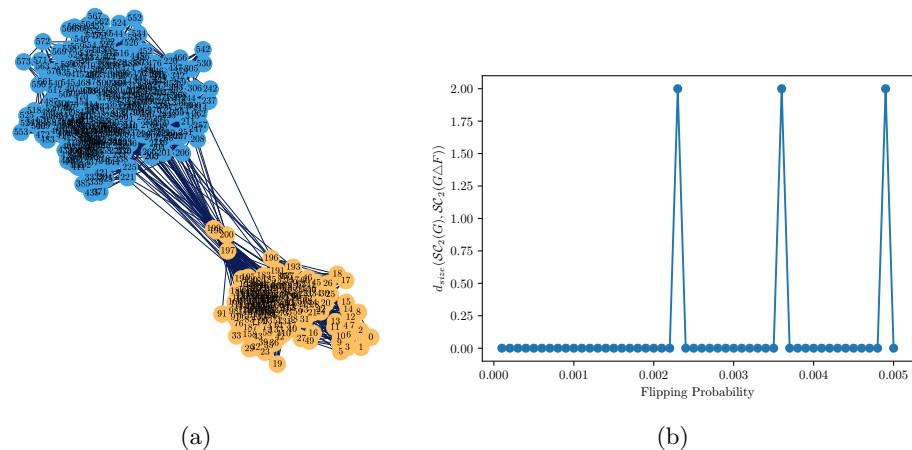

(a)  (b)

Figure 5.1: (a): The social network Facebook1684 obtained from SNAP after pruning. Each node was assigned a color based on the spectral clustering outcomes. (b): We generated 100 graphs from the first graph Facebook0 (Figure 1.1), and plotted the worst discrepancy $d_{\text{size}}$ between the outputs of the spectral clustering of the original and perturbed graphs for these 100 random runs.

a dense cluster of blue nodes. For ease of reference, these clusters will be subsequently named Cluster A, Cluster B, and Cluster C, respectively. Both Facebook0 and Facebook1684 have the attributes necessary for Assumption 2.12.

Our main theorem ensures that the clustering outcomes remain mostly consistent when edges are flipped with a probability $p < \frac{\log n}{10n}$. The upper bound is about 0.004 for Facebook0 and about 0.001 for Facebook1684. We examine $p \in \{0.0001q : 1 \leq q \leq 50\}$. For each probability $p$ and graph, we create 100 random graphs $F$ with the given probability. Note that the original graph is represented by $G$. We then compute the difference between the clustering results of $G$ (represented by $\mathcal{SC}_2(G)$) and that of $G \triangle F$ (represented by $\mathcal{SC}_2(G \triangle F)$).

The chart in Figure 5.1b shows the result we obtain from the first graph. The chart demonstrates the difference between the clustering outputs, represented as $d_{\text{size}}(\mathcal{SC}_2(G), \mathcal{SC}_2(G \triangle F))$. The values shown represent the worst-case results across 100 randomly generated graphs for each probability setting. We present the worst outcome among the 100 trials to highlight that, even in the least favorable scenario, the distance remains small. This demonstrates the robustness of the method.

Figure 5.1b reveals that, across all considered probabilities, the clustering outcomes remain consistent in every random graph. In each instance, when comparing the original graph to the graph with flipped edges, a minimum of 116 nodes are assigned to the same clusters. Only a maximum of four nodes out of 120 experience a change in their cluster placement.

For the second graph, the result is even more robust. For all the probabilities we have conducted the experiment, there were no change in the clustering results by the edge flipping. These two experiments suggest that the clustering results exhibit strong resilience to edge flipping.

## 5.1 Results on Larger Flipping Probability

In Figure 5.2, the flipping probability is raised above the level outlined in Assumption 2.12. Our experiments with the network derived from the Facebook0 network demonstrate that clustering outcomes remain stable as long as the flipping probability does not exceed 0.15.

In contrast, with the network derived from Facebook1684, the stability of the results is preserved only when the flipping probability remains under 0.04. Utilizing spectral clustering on the original graph divides it into two segments: one combining clusters A and B, and another comprising cluster C. However, exceeding a flipping probability of 0.04 occasionally alters the spectral clustering outcome to one group consisting of

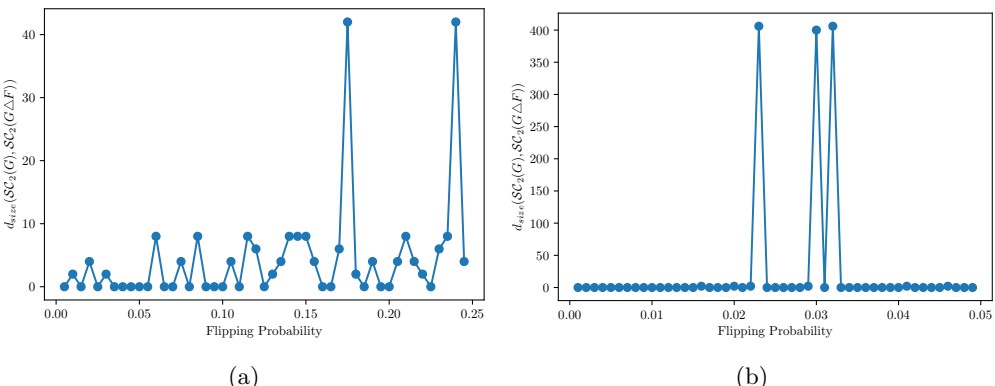

Figure 5.2: The robustness results of the social networks upon the introduction of a flipping probability that exceeds the value specified in Assumption 2.12 (a) for Facebook0 network and (b) for Facebook1684 network.

cluster A and another combining clusters B and C. This variation seems reasonable, as the latter grouping also yields a low conductance.

Section 4 demonstrates that in certain networks, clustering outcomes are unstable when the probability exceeds the level specified in Assumption 2.12. However, our experiments indicate that this threshold may be higher for particular graph types. In future work, we plan to develop theoretical results for these specific graphs.

### 5.2 Average Distances

While the maximum distances across the 100 iterations underscore the robustness of the algorithm, we also present the average distances in Figure 5.3 to offer a more comprehensive assessment. The plots reveal that, although the worst-case distance can be large, the average distance remains relatively small even under high flipping probabilities. This suggests that spectral clustering with randomized response maintains robustness on average, even when the flipping probability exceeds the theoretical bound established in our analysis.

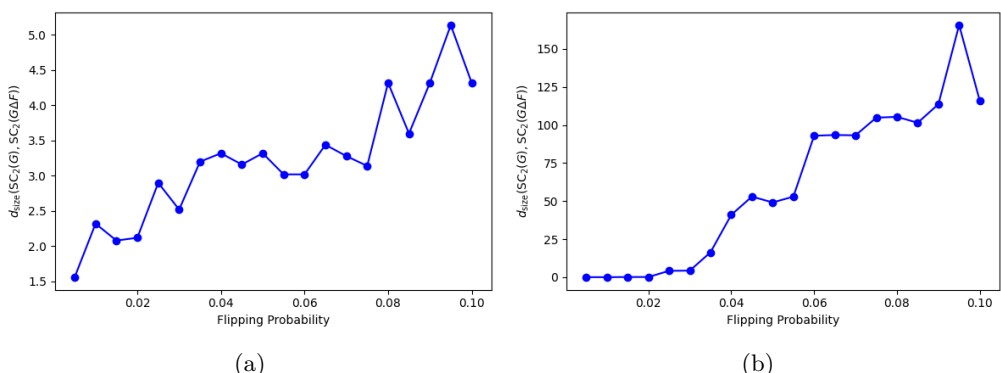

Figure 5.3: The average $d_{\text{size}}$ results of the social networks upon the introduction of a flipping probability that exceeds the value specified in Assumption 2.12 for (a) the Facebook0 network and (b) the Facebook1684 network.

## 6 Concluding Remarks

In this manuscript, we demonstrate and empirically verify that under some assumptions, the spectral clustering algorithm is robust under the randomized response method. While our primary objective is its use

in local differential privacy, our validation also confirms the robustness of spectral clustering against social networks containing inaccurate adjacency information. We demonstrate that the outcomes are robust when $p < \log n/10n$, but also acknowledge that the results can undergo significant alterations for larger $p$ values. This occurs because randomized response introduces an excessive number of edges to the graph in such cases. We are aiming to examine the robustness of other local (approximate) differential privacy approaches (e.g., as in Adhikari et al. (2020)) that do not add as many edges as the randomized response method.

Although in Peng & Yoshida (2020), there are results for spectral clustering with $k$ clusters, we cannot use ideas from those results in this work. Indeed, because we also consider edge addition, we have to demonstrate many additional theoretical results including Lemma 2.4 and Lemma 3.3. These analyses cannot be directly extended to the case when $k > 2$. We, anyway, believe that such an extension would be interesting future work.

## Acknowledgements

We sincerely thank the anonymous reviewers for their valuable feedback and Prof. Kamalika Choudhury, the action editor, for their kind and thoughtful handling of our manuscript. Sayan Mukherjee is supported by JSPS KAKENHI Grant Number 24K22830, and the Center of Innovations in Sustainable Quantum AI (JST Grant Number JPMJPF2221). Vorapong Suppakitpaisarn is supported by KAKENHI Grants JP21H05845, JP23H04377, and JP25K00369, as well as JST NEXUS Grant Number Y2024L0906031.

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
