# OpenReview forum: "Local Differential Privacy-Preserving Spectral Clustering for General Graphs"
_TMLR — Accepted by TMLR_

### Review · Reviewer_HF5H · 2025-02-27

**Summary Of Contributions:**

The paper contributes three primary results.

First, a novel theorem demonstrating that a large class of graphs which are "spectral clustering friendly" are able to be again clustered when the graph is released using randomized response for each edge with probability p = \frac{\log n}{n}. This goes beyond prior work which only provides similar guarantees for graphs in the stochastic block model (SBM).

Second, they provide evidence that graph clustering is not robust to edge flipping probability of higher than \frac{\log n}{n} by constructing a hard graph family.

Third, they empirically evaluate spectral clustering techniques on real world graphs, evaluating the effect of p on the clustering error.

**Audience:**

Yes

**Broader Impact Concerns:**

No broader impact statement is necessary.

**Claims And Evidence:**

No

**Requested Changes:**

The argument in Section 4 is too high-level, and could benefit from a formal theorem statement. In particular, it seems like the high-level argument could hold for any value of p, and it needs to be formally argued what happens when p suddenly becomes \frac{\log n}{n}.

The experimental graphs are too noisy, and while I think the high level trends are valid, I have concerns about reproducibility. Consider running each value of p 100 times and plotting the average clustering distance. This would show a nice, smooth upwards trend in the clustering error against p.

In Section 3, it would be nice to spend some time interpreting the obtained result, specifically for which graphs we can expect \eta(G) to be small (possibly in terms of the sparsity of the graph). Furthermore, it would be helpful to make a comparison here to prior work---while that work makes much stronger assumptions about the graph, is it possible to apply Theorem 3.1 to SBM graphs and recover the prior results?

**Strengths And Weaknesses:**

The strength of the paper lies in its first and third contributions. These results are the first of their kind to provide evidence for the performance of randomized response for graphs outside the SBM. The first theorem makes much more mild assumptions on the input graph than it being drawn from the SBM, and has some interesting proof techniques. The experimental design is well-founded.

A primary weakness of the paper is that the results are a bit less significant from a privacy point of view. They are demonstrating that a privacy guarantee of \epsilon > \log(n) is necessary in order to be able to have utility from the algorithm. However, this would be considered quite low privacy in the broader literature. This is also unappealing given that the prior work shows that clustering is possible for small values of epsilon for SBM graphs using better algorithms.

A second weakness is concerns about rigor. The clustering instability of the hard graph family seems to only hold in expectation, and I think a formal proof of instability is required. The argument in its current state is quite high-level, and it does make it clear what is special about the choice p = \frac{\log n}{n} for instability. Furthermore, the experimental data could be made less noisy.

---

> ### Author Response · Authors · 2025-03-12
> **Thank you very much for your kindness in considering our paper and your comments.**
>
> Thank you for your thoughtful consideration of our paper and your valuable comments. We have addressed your questions as follows.
>
> >The argument in Section 4 is too high-level, and could benefit from a formal theorem statement...
>
> Indeed, our treatment in Section 4 is very high-level. We will make the argument in Section 4 more formal and thorough in our next revision as follows.
>
> First, we would like to point out that the choices of the probabilities $1/10n$ and $\log n/n$ stem from the assumptions of $\lambda_2\ge 1/10$ and $\lambda_3\ge 10\log n$. These values are chosen in such a way that the left graph satisfies Assumption 2.10 in expectation. Moreover, concentration results from [Chung-Radcliffe, Electronic J. Combin. 2011] and [Lu-Peng, Electronic J. Combin. 2014] show that the Laplacian spectrum of our random construction is close to the spectrum of the expected Laplacian matrix, implying that our construction satisfies the eigenvalue conditions with high probability.
>
> Next, we will formally make the following statement about the instability of the sparsest cut:
>
> Theorem: The sparsest cut of the left graph is $(A, B \cup C)$. On the other hand, after the bit flipping with probability $p \gg \log n / n$, for any $S \subseteq B$, $\alpha(A \cup S) > \alpha(A)$ with probability close to 1/2.
>
> The proof sketch for the second part of this theorem is as follows.
> First, $e(A\cup S, (B-S)\cup C)$ and $e(A, B\cup C)$ both follow binomial distributions with probabilities $p-o(p)$. Thus, for large $n$, $\alpha(A\cup S)$ and $\alpha(A)$ will have distributions well-approximated by the normal distribution. However, $E[\alpha(A)] = E[\alpha(A\cup S)] = p-o(p)$, implying that $Pr[\alpha(A\cup S)\le \alpha (A)] \approx 0.5$ for any $S\subseteq B$.
>
> This theorem implies that the probability that $A$ remains the sparsest cut in the graph after flipping is exponentially small. We will make this argument formal in the revised version.
>
> >The experimental graphs are too noisy, and while I think the high level trends are valid, I have concerns about reproducibility...
>
> In our theoretical analysis, we argue that spectral clustering provides robust results against bit flipping with almost probability. To support this claim experimentally, we aim to demonstrate that, across 100 iterations, no single iteration produces a significantly large clustering distance. This would indicate robustness with high probability. For this reason, in the previous version of our manuscript, we chose to report the maximum clustering distance rather than the average.
> However, we acknowledge the reviewer's suggestion that the average clustering distance also offers valuable insight. Accordingly, we will include these results in the next version of the manuscript.
> Please find the results for the Facebook0 graph below.
> |Flipping Probability|Average Clustering Distance|
> |--------------------|--------------------------|
> |0.005|1.56|
> |0.01|2.32|
> |0.015|2.08|
> |0.02|2.12|
> |0.025|2.90|
> |0.03|2.52|
> |0.035|3.20|
> |0.04|3.32|
> |0.045|3.16|
> |0.05|3.32|
> |0.055|3.02|
> |0.06|3.02|
> |0.065|3.44|
> |0.07|3.28|
> |0.075|3.14|
> |0.08|4.32|
> |0.085|3.60|
> |0.09|4.32|
> |0.095|5.14|
> |0.1|4.32|
>
> Here is the result for the Facebook1684 graph.
> | Flipping Probability | Average Clustering Distance |
> |----------------------|----------------------------|
> |0.005|0.02|
> |0.01|0.0|
> |0.015|0.1|
> |0.02|0.12|
> |0.025|4.16|
> |0.03|4.30|
> |0.035|16.36|
> |0.04|40.92|
> |0.045|52.94|
> |0.05|49.1|
> |0.055|52.90|
> |0.06|92.90|
> |0.065|93.4|
> |0.07|93.18|
> |0.075|104.80|
> |0.08|105.32|
> |0.085|101.44|
> |0.09|113.64|
> |0.095|165.60|
> |0.1|115.74|
>
> >In Section 3, it would be nice to spend some time interpreting the obtained result, specifically for which graphs we can expect \eta(G) to be small (possibly in terms of the sparsity of the graph)...
>
> As pointed out in this question, indeed it is possible to use Theorem 3.1 to SBM graphs. We briefly mention this application in Remark 2.11, but would be happy to expand more in the next version.
>
> We would also like to mention that our assumption closely relates to that for spectral clustering analysis in [Peng, Sun, and Zanetti, ICML 2015].
> Let $\mu_1,\ldots, \mu_n$ be the normalized Laplacian eigenvalues. The authors showed in Eq. (1.3) that when $\mu_3\gg \phi(G)$, where $\phi(G)$ denotes the graph conductance, $G$ is well-clustered, and spectral clustering recovers these two clusters with high accuracy.
>
> In our work, we consider the unnormalized Laplacian eigenvalues $\lambda_1,\ldots, \lambda_n$, with the corresponding gap assumption $\lambda_3\gg n\alpha(G)$.
>
> By Cheeger’s inequality, $\eta(G)\ge c(n\alpha(G)/\lambda_3)^2$, so having small $\eta(G)$ is a stronger condition than the gap assumption in [Peng, Sun, Zanetti, ICML 2015]. This implies that graphs with small $\eta(G)$ are not only well-clustered, enabling accurate spectral clustering, but also robust to edge flipping.

---

### Review · Reviewer_ngYy · 2025-03-06

**Summary Of Contributions:**

Submission studies the problem of spectral graph clustering stability under local $\epsilon$-edge differential privacy constraints. In contrast to prior works, the authors analyze general graphs. They provide a high probability upper bound (depending on the spectral robustness of the graph) on the number of misclassified vertices for a flipped probability of $\mathcal{O}(\log n / n)$. In addition, they argue that a higher flipping probability of $\omega(\log n / n)$ may result in unstable private clustering for some graphs. Experiments on Facebook social networks support the theoretical results.

**Audience:**

Yes

**Broader Impact Concerns:**

No concerns for this submission

**Claims And Evidence:**

Yes

**Requested Changes:**

- Please provide references to justify claims about the properties of social networks. Specifically:

> “Social networks are not dense in practice.”
> “The second assumption derives from usual properties of social networks.”

Citing relevant literature would strengthen these statements.

- A discussion on the technical novelty of the proofs and techniques would be valuable. Clarifying whether the derivations build upon existing methods or introduce fundamentally new ideas would help contextualize the contributions.

### Minor Changes and Questions:
- A table comparing results to prior works would improve clarity and provide better context alongside the textual description.

- In Theorem 1.1, the phrase “small constant” $\eta(G)$ should be specified more concretely. As currently stated, $\eta(G) > 1$ would render the result meaningless since the number of misclassified vertices would exceed the total number of vertices $n$.

- The notation $\sim_p$ in Section 2.1 is unclear on first reading. Does it mean that each edge is sampled with probability $p$, or is it sampled uniformly among all subsets? The phrase “uniformly with probability $p$” is ambiguous and should be clarified.

- Since $\eta(G)$ plays a crucial role in the core result, defining its possible range and interpretation would be helpful for better understanding.

- The description of the first step of the spectral clustering algorithm lacks clarity. How is “reordering” performed? What is the computational complexity of the method? What is meant by “approximate eigenvector computation”? How does this approximation impact the clustering and any associated privacy guarantees?

- Lemma 2.6 relies on Lemma 3.5. For completeness, it would be useful to restate Lemma 3.5 (e.g., in the Appendix) and provide a more detailed proof.

- In Figure 4.1, what does the notation $K$ represent? A brief explanation in the caption or main text would clarify this.

**Strengths And Weaknesses:**

## Strengths

**S1.** The paper provides a comprehensive treatment of the problem via theoretical and experimental results. The writing is mostly clear, and the submission is well-structured by giving most of the necessary background so that it is possible for a person who is not an expert in this particular area to understand it.

**S2.** The theoretical analysis considers general graphs, a more realistic setting than the commonly assumed Stochastic Block Model (SBM), which seems a valuable contribution.

**S3.** Submission is quite accurate from a mathematical perspective. The definitions, assumptions, and necessary lemmas are provided in the main text with proofs. I did not note errors or inconsistencies. However, I did not verify all the steps of the main theorem.

## Weaknesses

**W1.** Applicability/generalizability of presented experimental results seems limited as the conclusions are based on 2 graphs. The evidence is also not always presented for all claims.

**W2.** Some definitions (e.g., in Section 2.1) and assumptions (e.g., about social network properties in Assumption 2.10) lack sufficient clarity. There are several requests for clarification in Section 4 related to notation.

**W3.** Bound in (1.1) looks restrictive as large $n$, i.e., $10^6$, results in $\epsilon > 6$, which seems a pretty weak DP guarantee in general. However, I know that for some ML applications, it can be more reasonable, but I do not know what is meaningful in this graph clustering with local DP context. Moreover, desirable small $\epsilon < 1$ seems unreachable.


### Minor questions

- I am also curious why pure DP is considered. Has approximate DP been investigated in such a context?
- It is a bit unclear why there is such a focus on this particular combination of clustering algorithm and DP mechanism. Why not other approaches?
- The reasoning in the first sentence of the commentary on the plausibility of Assumption 2.10 is not fully clear to me as Section 2.2 reasons the privacy budget to be lower-bounded by $\log 1/p$.
- Why random graphs $\mathcal{G}(n, p)$ appear in the proof for general graphs?
- What is $B$-$B$ edge? Does it mean edges between vertices from set $B$?

---

> ### Author Response · Authors · 2025-03-13
> **Thank you very much for your kindness in considering our paper and your comments.**
>
> > I am also curious why pure DP is considered...
>
> We believe that applying a non-private algorithm to the randomized response outputs is a standard approach to obtaining a locally differentially private algorithm. Therefore, in this work, we analyze the effect of applying spectral clustering to the randomized response results. Consequently, we achieve pure differential privacy from the randomized response mechanism.
>
> At the same time, we acknowledge that exploring approximate differential privacy for the graph clustering task under local differential privacy is an important direction. We plan to address this issue in our future work.
>
> > It is a bit unclear why there is such a focus on this particular combination...
>
> Randomized response is one of the most widely used mechanisms for achieving edge-level local differential privacy. To the best of our knowledge, no non-interactive method for graph local differential privacy exists without relying on this mechanism or its variants.
> Spectral clustering is also a commonly used method for graph clustering. We believe that it can be suitable for the randomized response result because it is robust against the random edge removal [Peng and Yoshida, KDD 2020].
>
> Indeed, other approaches for clustering and DP mechanisms would be very interesting to consider. However, our work tries to give insight into the problem of LDP graph clustering via spectral algorithms and prove theoretical guarantees on spectral clustering. Analysis of LDP versions of other clustering algorithms is ongoing work.
>
> We will emphasize the importance of the randomized response mechanism and spectral clustering in the next version of our manuscript.
>
> > The reasoning in the first sentence of the commentary on the plausibility of Assumption 2.10...
>
> We mean to say that if the privacy budget $O(\log n)$ has to be attained, $p$ must be in the order of $\Omega(1 / n)$. We have read the sentence again after we have obtained this comment, and we think that the sentence’s readability can be improved. We will do that in the next version of this paper.
>
> > What is $B$ - $B$ edge? ...
>
> Indeed, we mean the edges from $B$ to $B$. We realize that the writing in Section 4 needs to be more precise, and we will work on this in the updated version.
>
> > Why do random graphs appear in the proof for general graphs?
>
> The randomized response mechanism flips the relationship between all pairs of nodes in the graph with probability $p$. In our proof, we interpret this process as follows:
>
> (1) Constructing an Erdős-Rényi graph, where each pair of nodes is connected with probability $p$.
>
> (2) Flipping the edges of the original graph at positions where edges exist in the Erdős-Rényi graph generated in step (1).
>
> This interpretation explains why we consider the Erdős-Rényi graph in our proof.
>
> > What is $\sim_p$? ...
>
> For a big set $A$, by $S\sim_p A$ we mean that $S$ a subset of $A$ such that $Pr[x\in S]=p$ for every $x\in A$. We realize that this notation might be non-standard for the computer science audience, and we will clarify it in the next version.
>
> > In Theorem 1.1, the phrase “small constant” should be specified...
> > Since  $\eta(G)$ plays a crucial role in the core result, ...
>
> We would like to point out that as $\lambda_2/\lambda_3^2 < \frac 1{\lambda_3}$, typically $\eta(G)$ is going to be smaller than $\frac 1{\log n}$ for graphs satisfying our assumptions.
>
> We will add a discussion on $\eta(G)$ and its relationship with the gap requirement for correctness of spectral clustering as investigated by [Peng, Sun and Zanetti, ICML 2015, https://arxiv.org/pdf/1411.2021]. Please also refer to our answer to the comments of Reviewer HF5H regarding this relationship.
>
> > The description of the first step of the spectral clustering algorithm lacks clarity....
>
> We write the algorithm in a short form because of the page limitation and we follow a typical way of presenting the spectral clustering algorithm in the graph theory community.  As discussed at Line -6 of Page 4, by a “reordering”, we simply mean a permutation $\pi: V(G)\to V(G)$ such that $v_{\pi(1)}\le \cdots\le v_{\pi(n)}$, followed by a relabeling of the vertex $i$ with the label $\pi(i)$. The complexity of this step is $O(n\log n)$.
>
> In practice, computing eigenvectors via the Lancosz Algorithm is very precise, and it is what we use in our experimental results. However, we mentioned “approximate eigenvector computation” to emphasize that faster algorithms may perform spectral clustering more efficiently using a method such as [Adil and Saranurak, arXiv 2024].
>
> > In Figure 4.1, what does the notation $K$ represent? ...
>
> $K_n$ refers to a clique of size $n$. We did not define this in the previous version, as it is a standard concept for researchers working on graph algorithms. However, we now recognize the importance of providing a definition in the next version, as the readership of TMLR extends beyond this specific research community.

---

### Review · Reviewer_cShT · 2025-03-12

**Summary Of Contributions:**

This paper reasons about private clustering of n-vertex graphs with k=2 clusters (which additionally satisfy some other well-clustering guarantees.) In particular, they consider a previously studied local DP algorithm for producing a private copy of the input graph, and reason about the utility of running spectral clustering on this copy. While prior work in this area only considers graphs generated from the stochastic block model, this work considers a broader class of graphs, albeit still satisfying a number of assumptions on the clustering. They show that for this larger class of graphs, one must use a pretty high value for epsilon in order to preserve the desired utility of spectral clustering.

**Audience:**

Yes

**Broader Impact Concerns:**

An epsilon of log(n) can support extremely non private algorithms which I believe needs to be addressed in the manuscript and presentation if the work is accepted to TMLR.

**Claims And Evidence:**

Yes

**Requested Changes:**

Ways to strengthen the work:

Given that a large part of the motivation of the paper is to study more general social networks, it would have helped to have some discussion of the ways in which the studied graphs represent existing social networks better than those generated by SBMs. Especially since this work is restricted to the case of two clusters.

Minor comments:

It would be helpful for the definition of an individual to map to the definition of neighboring dataset. (The current introduction states that nodes are individuals, but then privacy guarantee considered in this work local-edge-DP.)

The introduction seems to imply that having a privacy budget $\epsilon$ is unique to local differential privacy, this should be corrected since it may confuse readers.

**Strengths And Weaknesses:**

Strengths:

The paper evaluates the behavior of the algorithm both analytically and experimentally.

Weaknesses:

An epsilon of log(n) can support extremely non private algorithms, yet the paper doesn't recommend against using such a high value for privacy, or justify why such a high epsilon still provides the effective privacy desired from the proposed mechanism.

---

> ### Author Response · Authors · 2025-03-13
> **Thank you very much for your kindness in considering our paper and your comments.**
>
> > An epsilon of log(n) can support extremely non private algorithms, yet the paper doesn't recommend against using such a high value for privacy, or justify why such a high epsilon still provides the effective privacy desired from the proposed mechanism.
>
> > An epsilon of log(n) can support extremely non private algorithms which I believe needs to be addressed in the manuscript and presentation if the work is accepted to TMLR.
>
> We agree with the reviewer that the value of $\epsilon$ in the order of $\Omega(\log n)$ could be considered too large. However, in Section 4 of our manuscript, we also demonstrate that it is not possible to attain the privacy budget of the order $o(\log n)$ for this algorithm. One might argue that having a positive result only for large epsilon is not impactful. However, we believe that a result matching the lower bound is still of interest to the TMLR audience, which supports our paper’s alignment with the acceptance criteria guidelines of TMLR.
>
> On the other hand, we agree with the reviewer that we should justify this privacy budget. We will do that in the next version of our manuscript.
>
> > Given that a large part of the motivation of the paper is to study more general social networks, it would have helped to have some discussion of the ways in which the studied graphs represent existing social networks better than those generated by SBMs. Especially since this work is restricted to the case of two clusters.
>
> Thank you very much for your comments. We agree with this point and will enhance the motivation in the next version of our paper. Specifically, we observed that none of the clusters in the published social networks on SNAP (SNAP: Stanford Network Analysis Project) exhibit a degree distribution that follows a mixture of two binomial distributions, which is a key property of SBMs.
>
> > It would be helpful for the definition of an individual to map to the definition of neighboring dataset. (The current introduction states that nodes are individuals, but then privacy guarantee considered in this work local-edge-DP.)
>
> Thank you very much for the comment. We will include the intuition why we need the local edge-DP in the next version of our paper. Indeed, in the local edge DP, we do not protect the information if an individual is participating in a database or protect the overall information of a particular user. We protect the information if there is an edge between two nodes.
>
> > The introduction seems to imply that having a privacy budget is unique to local differential privacy, this should be corrected since it may confuse readers.
>
> Thank you very much for pointing out this oversight on our part. We will rephrase the sentences in the first and the second paragraphs of the introduction and move the sentence regarding the privacy budget to the paragraph on DP.

---

> > ### Comment · Reviewer_cShT · 2025-03-13
> >
> > > We agree with the reviewer that the value of $\varepsilon$ in the order of log($n$) could be considered too large. However, in Section 4 of our manuscript, we also demonstrate that it is not possible to attain the privacy budget of the order $o($log($n$)$)$ for this algorithm. One might argue that having a positive result only for large epsilon is not impactful. However, we believe that a result matching the lower bound is still of interest to the TMLR audience, which supports our paper’s alignment with the acceptance criteria guidelines of TMLR.
> > >
> > >On the other hand, we agree with the reviewer that we should justify this privacy budget. We will do that in the next version of our manuscript.
> >
> > I agree that the impossibility of attaining a privacy budget of order $o($log($n$)$)$ for this algorithm is an interesting result. However, this does not imply impossibility for the problem overall. Furthermore, for epsilon values of this magnitude it is prudent to assume a lack of privacy in the absence of careful discussion and interpretation of the privacy guarantees of the algorithm being considered. I would strongly encourage the authors to address the practical implications of such high epsilon values in a broader impact statement.

---

> > > ### Author Response · Authors · 2025-03-14
> > > **Thank you very much for your kind reply and consideration.**
> > >
> > > Thank you for your thoughtful feedback. We agree with the reviewer about the importance of justifying the choice of privacy budget and will address this carefully in the revised manuscript.
> > >
> > > Regarding the broader impact statement, while our understanding is that TMLR guidelines primarily focus on ethical considerations involving potential risks of harm, we appreciate the reviewer's point about clearly communicating the practical implications of using large epsilon values. We will thoughtfully incorporate this discussion into our revision.

---

### Decision · Action_Editor_ssLw · 2025-04-17

**Recommendation:** Accept as is

**Comment:**

The results are sound, and supported by evidence. As a result the paper passes the TMLR acceptance bar.

**Audience:**

Audience is the section of the community that cares about differential privacy.

**Claims And Evidence:**

Overall this paper makes a verifiable claim and supports it with theoretical evidence. The claim overall appears sound.
Two reviewers have raised the issue that the privacy narrative surrounding this paper may not be as strong -- since O(log n)-DP offers very poor privacy. While this makes the paper not the most interesting, since the claims are correct and sound, it passes TMLR's bar for acceptance.